# Predicting Medications from Diagnostic Codes with Recurrent Neural Networks

**Jacek M. Bajor, Thomas A. Lasko**
Department of Biomedical Informatics
Vanderbilt University School of Medicine
Nashville, TN 37203, USA
{jacek.m.bajor,tom.lasko}@vanderbilt.edu

## Abstract

It is a surprising fact that electronic medical records are failing at one of their primary purposes, that of tracking the set of medications that the patient is actively taking. Studies estimate that up to 50% of such lists omit active drugs, and that up to 25% of all active medications do not appear on the appropriate patient list. Manual efforts to maintain these lists involve a great deal of tedious human labor, which could be reduced by computational tools to suggest likely missing or incorrect medications on a patient's list. We report here an application of recurrent neural networks to predict the likely therapeutic classes of medications that a patient is taking, given a sequence of the last 100 billing codes in their record. Our best model was a GRU that achieved high prediction accuracy (micro-averaged AUC 0.93, Label Ranking Loss 0.076), limited by hardware constraints on model size. Additionally, examining individual cases revealed that many of the predictions marked incorrect were likely to be examples of either omitted medications or omitted billing codes, supporting our assertion of a substantial number of errors and omissions in the data, and the likelihood of models such as these to help correct them.

## 1 Introduction

The idea of exploiting the large amounts of data captured in electronic medical records for both clinical care and secondary research holds great promise, but its potential is weakened by errors and omissions in those records (Safran et al., 2007; de Lusignan & van Weel, 2006). Among many other problems, accurately capturing the list of medications currently taken by a given patient is extremely challenging (Velo & Minuz, 2009). In one study, over 50% of electronic medication lists contained omissions (Caglar et al., 2011), and in another, 25% of all medications taken by patients were not recorded (Kaboli et al., 2004). Even medication lists provided by the patients themselves contain multiple errors and omissions (Green et al., 2010) .

Many efforts have been made to ensure the correctness of medication lists, most of them involving improved communication between patients and providers (Keogh et al., 2016), but these efforts have not yet been successful, and incorrect or incomplete medication documentation continues to be a source of error in computational medical research. In this work we attempt to identify likely errors and omissions in the record, predicting the set of active medications from the sequence of most recent disease-based billing codes in the record. Predictions from such a model could be used either in manual medication reconciliation (a common process undertaken to correct the medication record) or to provide a prior to other models, such as an NLP model attempting to extract medication use from the narrative clinical text.

Given the sequential nature of clinical data, we suspected that recurrent neural networks would be a good architecture for making these predictions. In this work we investigate this potential, comparing the performance of recurrent networks to that of similarly-configured feed forward networks.

The input for each case is a sequence of ICD-9 billing codes (Section 2.1), for which the model produces a single, multi-label prediction of the therapeutic classes (Section 3.1) of medications taken by the patient during the period of time covered by the billing code sequence.

This work is designed to test how well the complete set of medications a patient is actively taking at a given moment can be predicted by the sequence of diagnostic billing codes leading up to that moment, in the context of non-trivial label noise. It also explores whether sequence-oriented recursive neural nets can do a better job of that prediction than standard feed-forward networks.

## 2 BACKGROUND

### 2.1 MEDICAL BILLING CODES

Each time a patient has billable contact with the healthcare system, one or more date-stamped billing codes are attached to the patient record, indicating the medical conditions that are associated (or suspected to be associated) with the reason for the visit. While these codes are notoriously unreliable because they are only used for billing and not actual clinical practice (O'Malley et al., 2005), they are nevertheless useful in a research context (Bastarache & Denny, 2011; Denny et al., 2010), especially if they are used probabilistically (Lasko, 2014). In our institution, codes from the International Classification of Diseases, Ninth Revision (ICD-9) have historically been used, although we have recently transitioned to the tenth revision (ICD-10). For this project, we used ICD-9 codes.

The ICD-9 hierarchy consists of 21 chapters roughly corresponding to a single organ system or pathologic class (Appendix B). Leaf-level codes in that tree represent single diseases or disease subtypes. For this project, we used a subset of the two thousand most common leaf-level codes as our input data.

### 2.2 RECURRENT NEURAL NETWORKS AND VARIATIONS

Most of the ICLR community are very familiar with recurrent neural networks and their variations, but we include a conceptual description of them here for readers coming from other fields. More thorough descriptions are available elsewhere (Graves, 2012; Olah, 2015).

A recurrent neural network is a variation in which the output of one node on input $x_t$ loops around to become an input to another node on input $x_{t+1}$, allowing information to be preserved as it iterates over an input data sequence (Figure 1). They were introduced in the 1980s (Rumelhart et al., 1986), but achieved explosive popularity only recently, after the development of methods to more reliably capture long-term dependencies, which significantly improved their performance on sequence-to-sequence mapping (Hochreiter & Schmidhuber, 1997; Sutskever et al., 2014).

The basic RNN unit has a simple internal structure (Figure 2a). Output from the previous iteration $h_{t-1}$ and the next input in a sequence $x_t$ are both fed to the network on the next iteration. The Long Short-Term Memory configuration (LSTM) introduces new, more complex internal structure (Figure 2b) consisting of four neural network layers and a cell state ($c_t$), which is carried from one iteration to another. The additional layers form *forget*, *input* and *output* gates, which allow for the information to be forgotten (reset) or passed on to varying degrees.

The LSTM model and its variations are commonly used in applications where sequence and temporal data are involved, such as in image captioning (Vinyals et al., 2014), language translation (Sutskever et al., 2014), and speech recognition (Graves et al., 2013). In many cases LSTM models define the state of the art, such as with a recent conversational speech recognizer that (slightly) outperforms professional transcriptionists (Xiong et al., 2016).

A recent variation on the LSTM architecture is the Gated Recurrent Unit (GRU) (Cho et al., 2014), which introduces a single *update* gate in place of input and forget gates (Figure 2c). GRUs perform as well as or better than LSTMs in many cases (Chung et al., 2014; Jozefowicz et al., 2015), and have the additional advantage of a simpler structure.

In this work we try both an LSTM and a GRU on our learning problem.

### 2.3 RELATED WORK

Little research in the computational medical domain has used recurrent neural networks. The earliest example we are aware of is the use of an LSTM model that produced reasonable accuracy

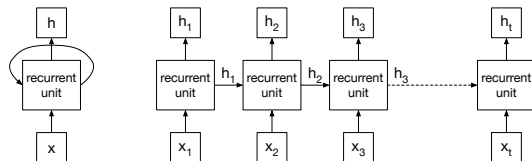

Figure 1: Simplified representation of a recurrent neural network (left) and an unrolled recurrent neural network (right). $x_i$ is a single element in an input sequence $x$, $h_i$ is an output after a single pass through the recurrent unit. Adapted from Olah (2015).

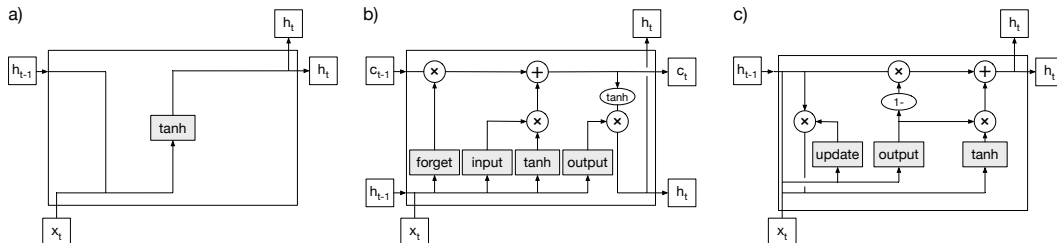

Figure 2: Architectures of (a) Simple RNN, (b) LSTM, and (c) GRU units. $x_t$: a single element in an input sequence being considered in the current iteration, $h_{t-1}, h_t$: the output from the previous and current iterations, $c_{t-1}, c_t$: the cell states of the previous and current iterations. Adapted from Olah (2015).

(micro-AUC 0.86) in a 128-dimensional multi-label prediction of diagnoses from regularly sampled, continuously-monitored, real-valued physiologic variables in an Intensive Care Unit setting. This was an interesting initial application, but it turned out to be only 0.001 better than the baseline classifier, which was a multi-layer perceptron with expert-designed features (Lipton et al., 2016). Given the dataset size (10,401 patient records) the lack of improvement may have been due to insufficient data to power accurate feature learning in the recurrent network.

Very recent work, contemporary with ours, used a GRU model with a semantic embedding in 32,787 patient records to predict the development of heart failure 3 - 6 months in the future, from medication orders and billing codes in an 18-month window. The model achieved respectable accuracy (0.88 AUC), and demonstrated a meaningful 0.05 AUC improvement over a deep feedforward network (Choi et al., 2016b).

Other recent work from the same group used a GRU model in a multi-label context to predict the medications, billing codes, and time of the next patient visit from a sequence of that same information for previous visits, using 263,706 patient records. It achieved a recall@30 of 72.4 for the task, an improvement of 20 over a single-hidden-layer MLP with 2000 units (Choi et al., 2016a). This is an example of using one of the strengths of a recurrent network - predicting the next element in a sequence. It contrasts with our work that exploits a different strength of recurrent networks - predicting a sequence or class that is semantically distinct from but parallel to the elements of the input sequence.

The closest work to ours from a medical domain perspective is a series of collaborative filter models (including co-occurrence counting, k-nearest neighbors, and logistic regression) that predict missing medications using a leave-one-drug-out evaluation design, with predictions based on the rest of the medications, ICD-9 billing codes, and demographic data. The models were trained and tested on data from 419 patients in three different clinics, with accuracy varying by clinic, as expected, but not appreciably by model. Most models ranked the missing drug in the top 10 results between 40 and 50% of the time, and ranked the therapeutic class of the drug in the top 10 results between 50 and 65% of the time.

Many aspects of our work can be found in these prior efforts, but none addresses our particular problem in the same way. Our work is unique in its learning problem of identifying all drugs a patient is likely to be taking, based only on the billing codes in the record. Like most others cited, we use recurrent neural networks in a multi-label predictive context, but in contrast to them we compare

to the most similar non-recurrent model we can construct, in order to evaluate the contribution of the temporal sequence information to the solution. Finally, we use one to four orders of magnitude more data (3.3 million instances, see Section 3.1) than these prior efforts, which we hope will give us a more realistic assessment of the various deep architectures we use on our problem.

# 3 EXPERIMENTS

## 3.1 DATA

Our source database was the deidentified mirror of Vanderbilt's Electronic Medical Record, which contains billing codes, medication histories, laboratory test results, narrative text and medical imaging data for over 2 million patients, reaching back nearly 30 years (Roden et al., 2008). We obtained IRB approval to use this data in this research.

For this experiment we filtered all records in our database to include only the top 1,000 most common medications and the top $m = 2000$ most common billing codes, which cover 99.5% of all medication occurrences and 85.1% of all billing code occurrences. We then included all records from the filtered data that had at least one medication occurrence and at least ten billing code occurrences. This resulted in 610,076 complete patient records, which we divided 80/5/15 into training, validation, and final test sets.

A data instance $d = \{E, T, y\}$ consisted of a sequence $E = \{e_1, \ldots, e_n\}$, of one-hot billing code vectors $e_i \in \{0, 1\}^m$ and their associated times $T = \{t_1, \ldots, t_n\}, t_i \in \mathbb{R}$ as input, and a multi-label vector $y \in \{0, 1\}^k$ of medication classes as the output target. The most recent $n = 100$ billing codes to a selected reference time point in a given patient record were collected into the input sequence $E$, and their occurrence times into $T$, zero padding if necessary. All medications that occurred during the time span of $T$ were then collected into the output vector $y$. Practice patterns change over time, so simply taking the most recent 100 codes for each patient could produce a biased result. To avoid this, we chose random reference points, stratified by medication. In other words, the reference points were randomly chosen from the occurrences of each medication in the entire dataset, up to 10,000 points per medication. This resulted in 3.3 million data instances, an average of 5.4 instances per patient record. Each patient's data was included in at most one of the training, validation, or test sets.

Because there are often many approximately equivalent medication choices for a given therapeutic purpose, we converted medication names to their therapeutic class (beta blocker, immunosuppressant, corticosteroid, etc.) as a synonym reduction step. This step also aggregated generic with brand names, as well as different formulations of the same active ingredient. For this task we used the Anatomical Chemical Classification System (ATC)[1], which is a multi-level ontology of medications, organized by both anatomic and therapeutic class. The top level is a broad categorization of medications (Appendix B), the bottom (fifth) level is individual medications, and we used the third level, which contains 287 therapeutic classes of the approximately appropriate abstraction level for our purpose. We used a publicly available mapping[2] to translate between our medication names and ATC codes, with manual mapping for the minority of medications that had no mapping entry. Our set of medications used $k = 182$ third-level ATC codes, rendering our output label a 182-element-long multi-label vector, in which an element is set $y_i = 1$ if a medication in that class appeared in the set of medications identified for that instance, $y_i = 0$ otherwise. Some medications mapped to more than one class, and we set $y_i = 1$ for all of them.

Our medication data was collected from structured order entry records and extracted using NLP (Xu et al., 2010) from mentions in the narrative text of a patient record that included the medication name, dose, route and frequency. As discussed above, we assumed (and our results demonstrate) that the medication data is incomplete, and our hope was that a model learned from a sufficiently large dataset will be robust to the missing data.

This configuration represents the input billing codes in a sequence, but the output medications as a multi-label vector. This is because ICD-9 codes are represented sequentially in our source data, but medications are not. They are represented as a list that changes over time in the record. The

---

[1]http://www.whocc.no/atc/structure_and_principles/
[2]https://www.nlm.nih.gov/research/umls/rxnorm/

usual goal of clinicians is to verify the list of medications at each visit, and if omissions or additions are indicated by the patient, to change the list to reflect that. But in the time-constrained reality of clinical practice, this reconciliation happens sporadically, and many clinicians are hesitant to change an entry on the medication list for which they were not the original prescriber, so the timing of the changes in the documentation do not reflect the timing of changes in reality. Therefore we are reduced to predicting a single multi-label vector, representing the medications that the patient probably took during the span of time represented by the input codes. (We actually did attempt some full sequence-to-sequence mappings, with various orderings of the medication sequences, but we did not achieve any promising results in that direction.)

## 3.2 CLASSIFIERS

Our main technical goal was to test the performance of recurrent neural networks on this sequence-centric prediction problem. To evaluate the specific gains provided by the recurrent architectures, we compare performance against a fully connected feed-forward network configured as similarly as possible to the recurrent networks, and (as baselines) a random forest and a constant-prevalence model. We discuss the specific configurations of these classifiers in this section.

### 3.2.1 RECURRENT NEURAL NETWORKS

We tested both LSTMs and GRUs in this experiment. We configured both architectures to first compute a semantic embedding $x_i \in \mathbb{R}^b$ of each input $e_i$ vector, before appending the times $t_i$ (Figure 3) and feeding the result to three layers of recurrent units. The final output from the last pass of recurrent unit is as a multi-label prediction for each candidate medication.

The optimal hyperparameters for the model were selected in the randomized parameter optimization (Bergstra & Bengio, 2012), with the embedding dimension $b = 32$, number of layers, and number of nodes optimized by a few trials of human-guided search. Other optimized parameters included the fraction of dropout (between layers, input gates and recurrent connections), and L1 and L2 regularization coefficients (final values are presented in Appendix A).

Both models were implemented using Keras (Chollet, 2015) and trained for 300 iterations using cross-entropy under the Adadelta optimizer (Zeiler, 2012).

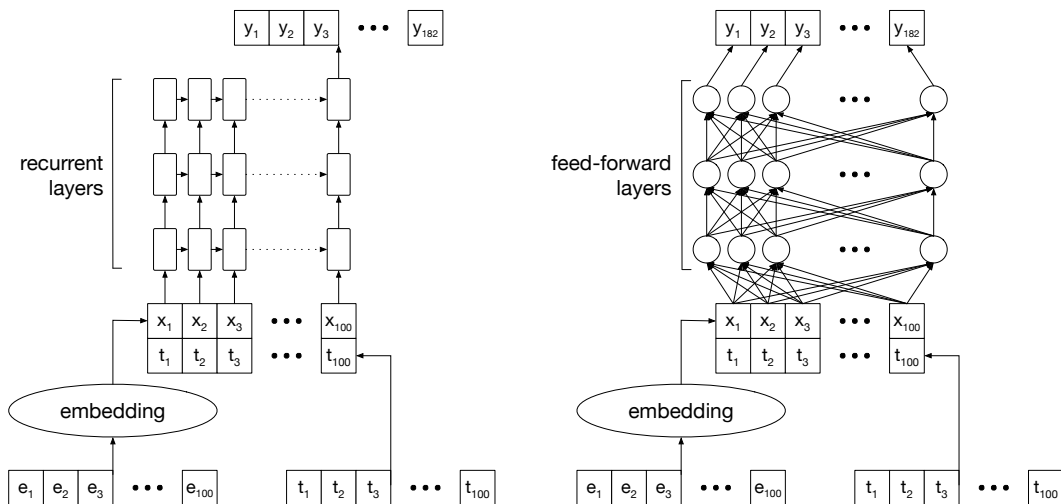

Figure 3: Recurrent (left) and feed-forward (right) neural network architectures. Arrows indicate the flow of information. Input for both models is sequence of billing code observations $e$ and sequence of corresponding timestamps $t$. A code observation $e_i$ passes through an embedding layer, producing an embedding vector $x_i$, which is then appended with time $t$. The processed matrix then passes through either recurrent layers or feed-forward layers. The output in both cases is a single vector $y$ of label probabilities.

### 3.2.2 FULLY CONNECTED NEURAL NETWORK

The fully connected network used as similar an architecture as possible to the recurrent networks, in an attempt to isolate the gain achieved from the recurrence property. Specifically, we used the same architecture for embedding and timestamp appending (Figure 3).

Hyperparameters were optimized using random search over the number of layers, number of nodes, dropout, activation function between layers, L1 and L2 regularization coefficients (Appendix A). (Surprisingly, the optimizer chose `tanh` over `ReLU` as the optimal activation function.)

The models were also implemented using Keras, and were trained using cross-entropy for 500 iterations under the Adadelta optimizer.

### 3.2.3 RANDOM FOREST

Because the random forest model is not easily structured to operate on sequences, we represented the input data as either binary occurrence vectors $v \in \{0,1\}^m$, or bag-of-codes vectors $w \in \mathbb{N}^m$ (counts of each code value in the sequence) rather than as sequences of codes with associated times. No embedding was used, because random forest code was not able to cope with the large size of the data in the (dense) embedded space.

Even in the (sparse) original space, the full dataset was too large for the random forest code, so we implemented it as an ensemble of ten independent forests, each trained on one tenth of the training data, and their average score used for test predictions.

Models were implemented using scikit-learn (Pedregosa et al., 2011) with parameters optimized under random search (Appendix A).

While other models could reasonably serve as a baseline for this work, we chose a random forest because they tend to perform well on widely varying datasets (Fernández-Delgado et al., 2014), they are efficient to train and test, and they don't require a huge effort to optimize (in order to produce a fair comparison).

## 3.3 CONSTANT-PREVALENCE MODEL

This minimum baseline model simply predicts the prevalence of each label for all instances. For example, if there were three possible medications, with prevalences of 0.3, 0.9, and 0.2, then the prediction of this model would be a constant $[0.3, 0.9, 0.2]$ for each instance. We include this model in order to mitigate the fact that while all of our evaluation measures are suitable for comparing models on the same data, some are not well suited for external comparison because they depend, for example, on the prevalence of positive labels (Section 3.4). By including this model we can at least establish a true minimum baseline for reference.

## 3.4 EVALUATION

Our main evaluation focused on the models, although we also performed a separate evaluation of the embedding.

### 3.4.1 MODELS

There are several possibilities for evaluation in a multi-label classification context (Sechidis et al., 2011; Zhang & Zhou, 2014). We chose micro-averaged area under the ROC curve (AUC) and label ranking loss as the primary methods of evaluation, because they treat each instance with equal weight, regardless of the nature of the positive labels for that instance. In other words, we wanted primary measures that did not give a scoring advantage to instances with either very many or very few positive labels, or that included very rare or very prevalent labels. Additionally, both of these measures appeal to us as intuitive extensions of the usual binary AUC, when seen from the perspective of a single instance. However, because these two measures don't reflect all aspects of multi-label prediction performance, we also include macro-averaged AUC, label ranking average precision and coverage error measures.

Micro-averaged AUC considers each of the multiple label predictions in each instance as either true or false, and then computes the binary AUC as if they all belonged to the same 2-class problem (Zhang & Zhou, 2014). In other words, micro-averaged AUC $A_\mu$ is:

$$A_\mu = \frac{\left|(x, x', l, l') : f(x, l) \geq f(x', l'), (x, l), \in \mathcal{S}, (x', l') \in \bar{\mathcal{S}}\right|}{|\mathcal{S}||\bar{\mathcal{S}}|}, \tag{1}$$

where $\mathcal{S} = \{(x, l) : l \in Y\}$ is the set of (instance, label) pairs with a positive label, and $Y = \{y_i : y_i = 1, i = 1 \ldots k\}$ is the set of positive labels for input $x$.

Label ranking loss $L_R$ gives the average fraction of all possible (positive, negative) label pairs for each instance in which the negative label has a higher score than the positive label (Tsoumakas et al., 2010):

$$L_R = \frac{1}{N} \sum_{j=1}^{N} \frac{1}{|Y^{(j)}||\overline{Y^{(j)}}|} \left|\left\{(l, l') : r^{(j)}(l) > r^{(j)}(l'), (l, l') \in Y^{(j)} \times \overline{Y^{(j)}} \right\}\right| \tag{2}$$

where the superscript $(j)$ refers to the $j$th test instance (of $N$ total instances) and $r(l)$ is the predicted rank of a label $l$.

Macro-averaged AUC can be thought of as averaging the AUC performance of several one-vs-all classifiers, one model for each label. It treats each model equally, regardless of the prevalence of positive labels for that model. This gives a score of 0.5 to the constant-prevalence model, at the cost of weighting instances differently in order to achieve that. This is in contrast to micro-averaged AUC, which can be thought of as averaging across instances rather than labels. It weighs each instance equally, at the cost of a 0.5 score no longer being the random-guessing baseline.

Label ranking average precision gives the mean fraction of correct positive labels among all positive labels with lower scores for each label. The coverage error function calculates the mean number of labels on the ranked list that are needed to cover all the positive labels of the sample. Both of these depend on the prevalence of positive labels in a test instance.

### 3.4.2 Embedding

We evaluated the embedding based on how strongly related in a clinical semantic sense the nearest neighbor to each code is (in the embedding space). A licensed physician manually annotated the list of all 2000 codes with its match category $m \in$ {strongly related, loosely related, unrelated}, and we computed the empirical marginal probability $P(m)$ of each category, the empirical conditional probability $P(m|d)$ of the match category given the nearest neighbor (Manhattan) distance $d$ and the empirical marginal probability $P(d)$. For comparison, we computed $P(m)$ under 100 random code pairings.

## 4 Results and Discussion

The GRU model had the top performance by all measures, although the LSTM was a close second (Table 1), a performance pattern consistent with previous reports (Chung et al., 2014). The deep neural net performance was about 0.01 worse in both measures, suggesting that the recurrent models were able to use the sequence information, but only to a small advantage over the most similar non-temporal architecture. However, we note that both RNNs' performance peaked at the top end of our tractable range for model size, while the feed-forward network peaked using a model about one third that size (Appendix A). Experimenting with the architecture, we found that increasing the number of nodes or layers for the feed-forward network increased training time but not performance. This suggests that the RNN performance was limited by the hardware available, and increasing the size of the model may further increase performance, and that the feed-forward network was limited by something else.

Both random forest models were weaker than the deep neural net, as might be expected from the need to resort to binary and bag-of-codes representations of the input data.

Table 1: Results of multi-label classification for each model. Baseline is the constant-prevalence model. Perfect is the best possible performance for our data under the given measure.

| Model | Micro-AUC | Label Ranking Loss | Macro-AUC | Label Ranking Avg. Precision | Coverage Error |
|---|---|---|---|---|---|
| GRU | **0.927** | **0.076** | **0.861** | **0.603** | **62.6** |
| LSTM | 0.926 | 0.077 | 0.859 | 0.600 | 63.0 |
| NN | 0.916 | 0.086 | 0.835 | 0.570 | 67.3 |
| RF (binary) | 0.903 | 0.102 | 0.804 | 0.523 | 73.7 |
| RF (counts) | 0.894 | 0.111 | 0.787 | 0.497 | 77.3 |
| Baseline | 0.828 | 0.172 | 0.500 | 0.355 | 97.2 |
| Perfect | 1.0 | 0.0 | 1.0 | 1.0 | 15.0 |

A natural question is what performance is *good enough* for clinical use. While there is little clinical experience with multi-label classifiers, we would generally expect clinicians using a *binary* classifier in an advisory role to find an AUC $\gtrsim 0.9$ to be useful, and AUC $\gtrsim 0.95$ to be very useful. An AUC difference of 0.01, and perhaps 0.005 are potentially noticeable in clinical use.

This 0.9/0.01 rule of thumb may loosely translate to our AUC variants, but it can directly translate to Label Ranking Loss $L_R$ (2). If we think of a single output prediction $\hat{y} \in [0,1]^k$ as a set of predictions for $k$ binary labels, then $1 - \text{AUC}$ for that set of predictions is equivalent to $L_R$ for the original instance $\hat{y}$. Therefore, values of $L_R \lesssim 0.1$ may be clinically useful, and $L_R \lesssim 0.05$ may be very useful.

Subjectively examining performance on 20 randomly selected cases, we find very good detailed predictions, but also evidence of both missing medications and missing billing codes. An example of a good set of detailed predictions is from a complex patient suffering from multiple myeloma (a type of cancer) with various complications. This patient was taking 26 medications, 24 of which had moderate to high probability predictions (Figure 4). (We have found by eyeball that a prediction cutoff of 0.2 gives a reasonable balance between sensitivity and specificity for our model.) In the other direction, only two of the high-prediction classes were not actually being taken, but those classes, along with several of the other moderately-predicted classes, are commonly used for cancer and are clinically reasonable for the case. (Details of this and the two cases below are in Appendix C).

A good example of missing medications is a case in which the record has multiple billing codes for both osteoporosis (which is very commonly treated with medication) and postablative hypothyroidism (a deliberately induced condition that is always treated with medication), but no medications of the appropriate classes were in the record. The GRU model predicted both of these classes, which the patient was almost surely taking.

A good example of either missing billing codes or discontinued medications that remain documented as active is a case in which the record has at least five years of data consisting only of codes for Parkinson's disease, but which lists medications for high cholesterol, hypertension, and other heart disease. The GRU model predicted a reasonable set of medications for Parkinson's disease and its complications, but did not predict the other medications that are not suggested by the record.

Given how easy it was to find cases with apparently missing codes and medications, we conclude that there is indeed a substantial amount of label noise in our data, and we therefore interpret our models' performance as lower bounds on the actual performance. We are encouraged that this kind of a model may actually be useful for identifying missing medications in the record, but of course a more thorough validation, and possibly a more accurate model, would be necessary before using in a clinical scenario. A definitive experiment would use off-line research, including reconciling information from various electronic and human sources to establish the ground truth of which medications were being taken on a particular day, but such efforts are labor intensive and expensive, and can only be conducted on a very small scale.

An interesting byproduct of these models is the semantic embedding of ICD-9 codes used in the recurrent networks (Figure 5). Transforming input to a semantic embedding is a common pre-

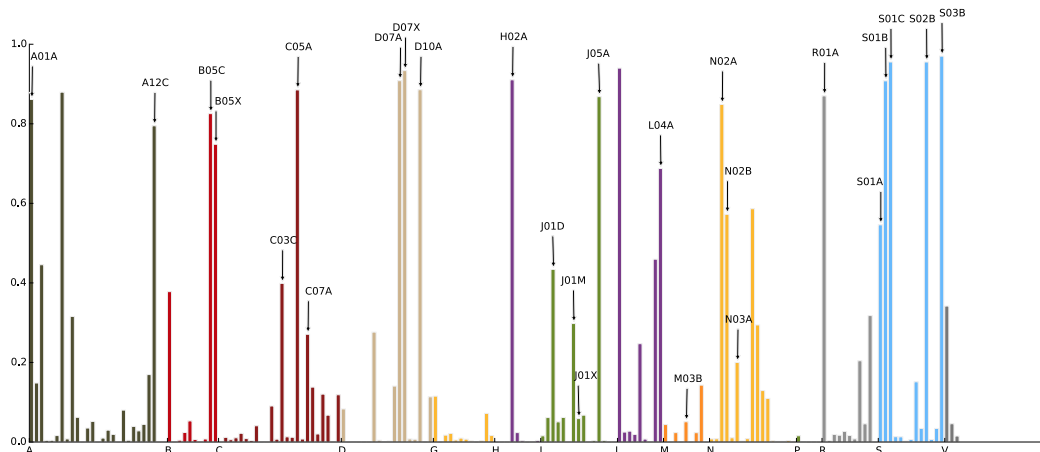

Figure 4: Medication predictions for a complicated patient. Each vertical bar represents the prediction for a single medication class, with the height of the bar representing the confidence of the prediction. Black labels with arrows indicate ATC therapeutic classes for medications the patient was actually taking. Colors and letters below the axis indicate organ system groups. More detail in Appendix C.

processing step to improve performance, but clearly the semantic understanding it provides to an algorithm can be useful beyond the immediate learning problem (Mikolov et al., 2013). Investigating the embedding learned in this experiment shows some generalizable potential, but it also reveals the need for further refinement before it can be truly useful. Specifically, while it's easy to find tight groups of ICD-9 codes that are strongly clinically related in our embedding, we also find groups for which we cannot see a meaningful clinical relationship.

For example, we see two groups of codes relating to kidney failure and diabetes mellitus, two classes of very prevalent disease (Figure 5, insets). In other iterations with different parameter settings, the kidney failure codes were even embedded in a sequence reflecting the natural progression of the disease, with the code for dialysis (an intensive treatment for end-stage kidney failure) embedded at the appropriate place. Interestingly, these were not the parameter settings that optimized overall prediction performance. In other settings, such as our performance-optimal setting, the sequence is close to the natural progression of the disease, but not quite identical. Nevertheless, this is an exciting result that suggests great potential.

Further evaluation of the embedding found that 49% of codes were strongly related semantically to their nearest neighbor, 10% were loosely related, and 41% unrelated. This fraction of strongly related nearest neighbors was lower than we had hoped, but much higher than expected by chance (Figure 6), and it definitely improved classification performance. Furthermore, it was obvious by inspection that in general, codes closer in the embedding were more semantically related than distant codes, but interestingly, the distance to the *nearest* such neighbor showed the opposite relationship — nearest neighbors that were very close were less likely to be semantically related than nearest neighbors that were far, and this trend is roughly linear across the full range of $d$ (Figure 6). So the sparser the points are in the embedded space, the more semantically related they are to their nearest neighbor, but the causal direction of that effect and the technical reason for it are beyond the scope of this initial work.

For this prediction problem, we settled on predicting the medications that occurred in the record during the same time span as the billing codes used. Originally, we intended to predict only the medications listed on the day of the reference point, but that turned out to greatly exacerbate the missing medication problem. After trying medications that fell on the reference day only, the week prior to the reference day, and the six months prior, our best performance both subjectively and objectively was achieved using the full time range of the input data.

While the performance of the recurrent networks was quite good, we believe it could be improved by including additional input data, such as laboratory test results, demographics, and perhaps vital

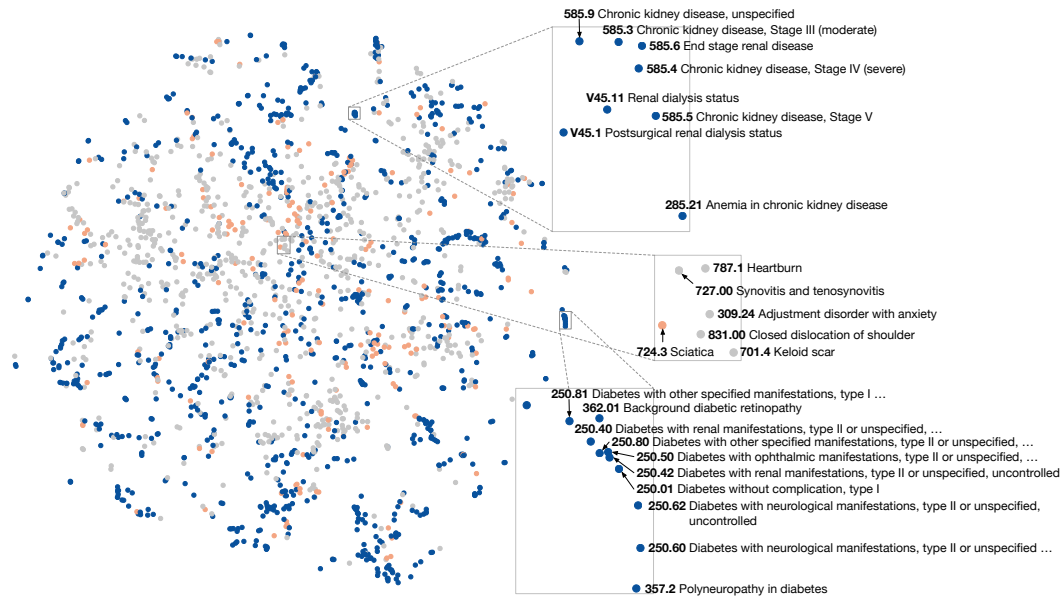

Figure 5: A t-SNE representation of our final embedding. The insets highlight two groups of codes (diabetes mellitus and kidney failure) that are strongly related clinically, and a third group that is not. Codes are colored by whether their nearest neighbor in the embedding space (which may be different from the nearest neighbor in this t-SNE space) is strongly related (blue), loosely related (orange), or unrelated (gray) from a clinical perspective.

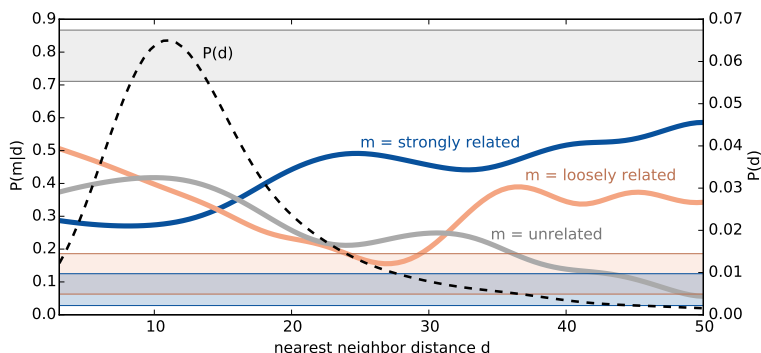

Figure 6: Semantic relatedness of nearest neighbors vs. the distance between them. Solid lines are the conditional probabilities $P(m|d)$ for the three values of $m$, dashed line is the marginal probability $P(d)$ of nearest neighbor distances $d$. Surprisingly, nearest neighbors that are farther away (but still the nearest neighbor) are more strongly related than nearest neighbors that are closer in the embedding space. Shaded regions, colored to correspond to the three values of $m$, are the 95% CI for empirically estimated $P(m)$ under random pairings, and represent the expected null result.

signs. We also suspect that if we can devise a way to convert our medication data into reliably-ordered sequences, we can more fully exploit the strengths of recurrent networks for medication prediction. We look forward to trying these and other variations in future work.

## ACKNOWLEDGMENTS

This work was funded by grants from the Edward Mallinckrodt, Jr. Foundation and the National Institutes of Health R21LM011664 and R01EB020666. Clinical data was provided by the Vanderbilt Synthetic Derivative, which is supported by institutional funding and by the Vanderbilt CTSA grant ULTR000445.

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

APPENDIX A.

This appendix lists the optimized parameters for the different models. Except where noted, parameters were optimized under random search.

Recurrent Neural Network Models: (parameters marked with an asterisk were optimized with human-guided search.)

| Parameter | Model | |
|---|---|---|
| | GRU | LSTM |
| Dropout for input gates | 0.1 | 0.25 |
| Dropout for recurrent connections | 0.75 | 0.75 |
| L1 applied to the input weights matrices | 0 | 0 |
| L1 applied to the recurrent weights matrices | 0 | 0 |
| L2 applied to the input weights matrices | 0.0001 | 0.0001 |
| L2 applied to the recurrent weights matrices | 0.0001 | 0.001 |
| L2 applied to the output layer's weights matrices | 0.0001 | 0.001 |
| Dropout before the output layer | 0.5 | 0.5 |
| *Number of recurrent layers | 3 | 3 |
| *Number of nodes in recurrent units | 400 | 400 |

Feed Forward Neural Network Model:

| Parameter | Value |
|---|---|
| Dropout before the output layer | 0.1 |
| Dropout between feed-forward layers | 0.1 |
| Number of feed-forward layers | 3 |
| Activation function between feed-forward layers | tanh |
| Number of nodes in feed-forward layers | 128 |

Random Forest Model (binary input):

| Parameter | Value |
|---|---|
| Number of estimators | 800 |
| Ratio of features to consider when looking for the best split | 0.4666 |
| Minimum number of samples required to split an internal node | 87 |
| Minimum number of samples required to be at a leaf node | 3 |
| The function to measure the quality of a split | entropy |

APPENDIX B.

This appendix lists the top level classes for International Statistical Classification of Diseases and Related Health Problems, Ninth Revision (ICD-9) and Anatomical Chemical Classification System (ATC).

ICD-9 chapters.

| Code range | Description |
|---|---|
| 001-139 | Infectious and parasitic diseases |
| 140-239 | Neoplasms |
| 240-279 | Endocrine, nutritional and metabolic diseases, and immunity disorders |
| 280-289 | Diseases of the blood and blood-forming organs |
| 290-319 | Mental disorders |
| 320-359 | Diseases of the nervous system |
| 360-389 | Diseases of the sense organs |
| 390-459 | Diseases of the circulatory system |
| 460-519 | Diseases of the respiratory system |
| 520-579 | Diseases of the digestive system |
| 580-629 | Diseases of the genitourinary system |
| 630-679 | Complications of pregnancy, childbirth, and the puerperium |
| 680-709 | Diseases of the skin and subcutaneous tissue |
| 710-739 | Diseases of the musculoskeletal system and connective tissue |
| 740-759 | Congenital anomalies |
| 760-779 | Certain conditions originating in the perinatal period |
| 780-799 | Symptoms, signs, and ill-defined conditions |
| 800-999 | Injury and poisoning |
| V01-V91 | Supplementary - factors influencing health status and contact with health services |
| E000-E999 | Supplementary - external causes of injury and poisoning |

Top level groups ATC codes and their corresponding colors used in Figure 4 and Appendix C.

| Code | Contents | Color |
|---|---|---|
| A | Alimentary tract and metabolism | 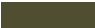 |
| B | Blood and blood forming organs |  |
| C | Cardiovascular system |  |
| D | Dermatologicals |  |
| G | Genito-urinary system and sex hormones |  |
| H | Systemic hormonal preparations, excluding sex hormones and insulins |  |
| J | Antiinfectives for systemic use |  |
| L | Antineoplastic and immunomodulating agents |  |
| M | Musculo-skeletal system |  |
| N | Nervous system |  |
| P | Antiparasitic products, insecticides and repellents |  |
| R | Respiratory system |  |
| S | Sensory organs |  |
| V | Various |  |

## APPENDIX C.

This appendix presents results from three illustrative cases from the dozen cases randomly selected for individual evaluation.

### CASE 1.

| ICD-9 code | Code description | Time estimate (ago) |
|---|---|---|
| 203.00 | Multiple myeloma, without mention of having achieved remission | 4.8 months ago |
| 273.1 | Monoclonal paraproteinemia | 4.8 months ago |
| 285.9 | Anemia, unspecified | 4.8 months ago |
| 276.50 | Volume depletion, unspecified | 4.8 months ago |
| 733.00 | Osteoporosis, unspecified | 4.8 months ago |
| 203.00 | Multiple myeloma, without mention of having achieved remission | 4.8 months ago |
| 203.00 | Multiple myeloma, without mention of having achieved remission | 2.9 months ago |
| 203.01 | Multiple myeloma, in remission | 2.9 months ago |
| 273.1 | Monoclonal paraproteinemia | 2.9 months ago |
| 273.1 | Monoclonal paraproteinemia | 1.6 months ago |
| 279.3 | Unspecified immunity deficiency | 1.6 months ago |
| 203.00 | Multiple myeloma, without mention of having achieved remission | 1.6 months ago |
| 781.2 | Abnormality of gait | 3.7 weeks ago |
| 203.00 | Multiple myeloma, without mention of having achieved remission | 3.7 weeks ago |
| 401.9 | Unspecified essential hypertension | 3.7 weeks ago |
| V12.54 | Personal history of transient ischemic attack (TIA), and cerebral infarction without residual deficits | 3.7 weeks ago |
| 794.31 | Nonspecific abnormal electrocardiogram [ECG] [EKG] | 3.7 weeks ago |
| 786.09 | Other respiratory abnormalities | 3.7 weeks ago |
| 273.1 | Monoclonal paraproteinemia | 3.7 weeks ago |
| 203.00 | Multiple myeloma, without mention of having achieved remission | 3.6 weeks ago |
| V58.69 | Long-term (current) use of other medications | 3.6 weeks ago |
| 794.31 | Nonspecific abnormal electrocardiogram [ECG] [EKG] | 3.4 weeks ago |
| 203.00 | Multiple myeloma, without mention of having achieved remission | 4 days ago |
| V42.82 | Peripheral stem cells replaced by transplant | 4 days ago |
| 203.01 | Multiple myeloma, in remission | 3 days ago |
| 38.97 | Central venous catheter placement with guidance | 3 days ago |
| V42.82 | Peripheral stem cells replaced by transplant | 3 days ago |
| V58.81 | Fitting and adjustment of vascular catheter | 3 days ago |
| 203.00 | Multiple myeloma, without mention of having achieved remission | 3 days ago |
| V42.82 | Peripheral stem cells replaced by transplant | 2 days ago |
| 203.01 | Multiple myeloma, in remission | 2 days ago |
| 203.00 | Multiple myeloma, without mention of having achieved remission | 1 day ago |
| V42.82 | Peripheral stem cells replaced by transplant | 1 day ago |
| 203.00 | Multiple myeloma, without mention of having achieved remission | now |
| V42.82 | Peripheral stem cells replaced by transplant | now |

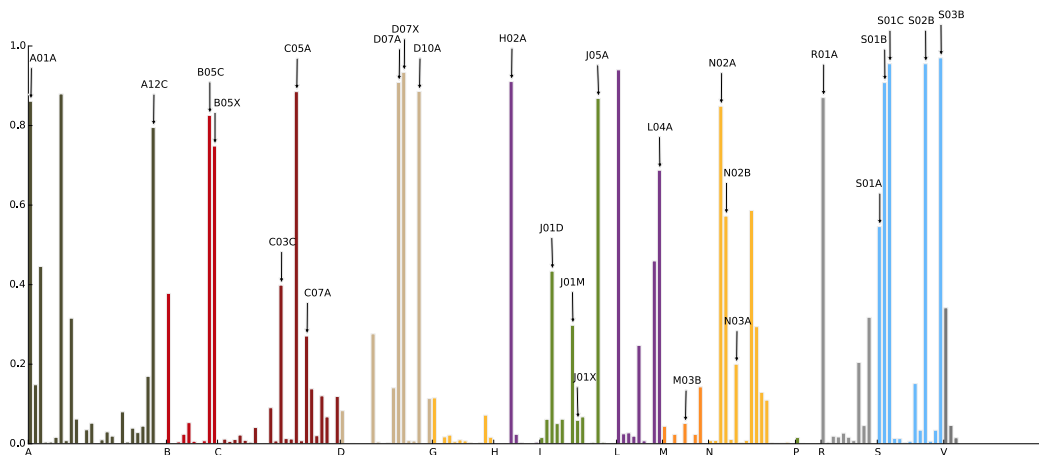

Medication predictions for a complicated patient. Each vertical bar represents the prediction for a single medication class, with the height of the bar representing the confidence of the prediction. Black labels above arrows indicate ATC therapeutic classes for medications the patient was actually taking. Colors and letters below the axis indicate high-level therapeutic class groups.

Predicted vs. actual medication classes for the patient in Case 1. The four-character sequence in the first and fourth columns is the ATC code for the medication therapeutic class, and an asterisk in the first column indicates that the predicted medication is in the actual medication list. Probabilities listed are the model predictions for the listed therapeutic class. In the predicted medications column, all predictions with probability at least 0.2 are listed.

| | Top predictions | Prob. | | True labels | Prob. |
|---|---|---|---|---|---|
| S03B* | Corticosteroids | 97.01% | S03B | Corticosteroids | 97.01% |
| S01C* | Antiinflammatory agents and antiinfectives in combination | 95.54% | S01C | Antiinflammatory agents and antiinfectives in combination | 95.54% |
| S02B* | Corticosteroids | 95.54% | S02B | Corticosteroids | 95.54% |
| L01A | Alkylating agents | 94.00% | D07X | Corticosteroids, other combinations | 93.37% |
| D07X* | Corticosteroids, other combinations | 93.37% | H02A | Corticosteroids for systemic use, plain | 91.06% |
| H02A* | Corticosteroids for systemic use, plain | 91.06% | D07A | Corticosteroids, plain | 90.83% |
| D07A* | Corticosteroids, plain | 90.83% | S01B | Antiinflammatory agents | 90.79% |
| S01B* | Antiinflammatory agents | 90.79% | D10A | Anti-acne preparations for topical use | 88.56% |
| D10A* | Anti-acne preparations for topical use | 88.56% | C05A | Agents for treatment of hemorrhoids and anal fissures for topical use | 88.52% |
| C05A* | Agents for treatment of hemorrhoids and anal fissures for topical use | 88.52% | R01A | Decongestants and other nasal preparations for topical use | 87.02% |
| A04A | Antiemetics and antinauseants | 87.95% | J05A | Direct acting antivirals | 86.83% |
| R01A* | Decongestants and other nasal preparations for topical use | 87.02% | A01A | Stomatological preparations | 86.11% |
| J05A* | Direct acting antivirals | 86.8% | N02A | Opioids | 84.86% |
| A01A* | Stomatological preparations | 86.11% | B05C | Irrigating solutions | 82.56% |
| N02A* | Opioids | 84.86% | A12C | Other mineral supplements | 79.50% |
| B05C* | Irrigating solutions | 82.56% | B05X | I.V. solution additives | 74.84% |
| A12C* | Other mineral supplements | 79.50% | L04A | Immunosuppressants | 68.76% |
| B05X* | I.v. solution additives | 74.84% | N02B | Other analgesics and antipyretics | 57.24% |
| L04A* | Immunosuppressants | 68.76% | S01A | Antiinfectives | 54.59% |
| N05A | Antipsychotics | 58.64% | J01D | Other beta-lactam antibacterials | 43.40% |
| N02B* | Other analgesics and antipyretics | 57.24% | C03C | High-ceiling diuretics | 39.88% |
| S01A* | Antiinfectives | 54.59% | J01M | Quinolone antibacterials | 29.78% |
| L03A | Immunostimulants | 45.96% | C07A | Beta blocking agents | 27.08% |
| A02B | Drugs for peptic ulcer and gastro-oesophageal reflux disease | 44.56% | | | |
| J01D* | Other beta-lactam antibacterials | 43.40% | N03A | Antiepileptics | 20.00% |
| C03C* | High-ceiling diuretics | 39.88% | J01X | Other antibacterials | 5.88% |
| B01A | Antithrombotic agents | 37.80% | M03B | Muscle relaxants, centrally acting agents | 5.09% |
| V03A | All other therapeutic products | 34.18% | | | |
| R06A | Antihistamines for systemic use | 31.78% | | | |
| A06A | Drugs for constipation | 31.57% | | | |
| J01M* | Quinolone antibacterials | 29.78% | | | |
| N05B | Anxiolytics | 29.42% | | | |
| D04A | Antipruritics, incl. antihistamines, anesthetics, etc. | 27.62% | | | |
| C07A* | Beta blocking agents | 27.08% | | | |
| L01X | Other antineoplastic agents | 24.72% | | | |
| R05C | Expectorants, excl. combinations with cough suppressants | 20.43% | | | |
| N03A* | Antiepileptics | 20.00% | | | |

CASE 2.

| ICD-9 code | Code description | Time estimate (ago) |
|---|---|---|
| 735.4 | Other hammer toe (acquired) | 2.4 years ago |
| 729.5 | Pain in limb | 2.4 years ago |
| 244.1 | Other postablative hypothyroidism | 1.5 years ago |
| 285.9 | Anemia, unspecified | 1.5 years ago |
| 244.1 | Other postablative hypothyroidism | 1.2 years ago |
| 244.1 | Other postablative hypothyroidism | 11.5 months ago |
| 733.00 | Osteoporosis, unspecified | 11.5 months ago |
| 733.01 | Senile osteoporosis | 7.7 months ago |
| 268.9 | Unspecified vitamin D deficiency | 7.7 months ago |
| 729.5 | Pain in limb | 7.7 months ago |
| 174.9 | Malignant neoplasm of breast (female), unspecified | 7.7 months ago |
| 722.52 | Degeneration of lumbar or lumbosacral intervertebral disc | 7.7 months ago |
| 279.3 | Unspecified immunity deficiency | 7.7 months ago |
| 733.01 | Senile osteoporosis | 6.4 months ago |
| 733.01 | Senile osteoporosis | 6.2 months ago |
| 244.1 | Other postablative hypothyroidism | 6.0 months ago |
| 401.1 | Benign essential hypertension | 6.0 months ago |
| V58.69 | Long-term (current) use of other medications | 1.9 weeks ago |
| 733.01 | Senile osteoporosis | now |
| 244.1 | Other postablative hypothyroidism | now |
| V58.69 | Long-term (current) use of other medications | now |

Predicted vs. actual medication classes for Case 2. Table structure as in case 1.

| | Top predictions | Prob. | | True labels | Prob. |
|---|---|---|---|---|---|
| M05B | Drugs affecting bone structure and mineralization | 88.18% | A11C | Vitamin a and d, incl. combinations of the two | 39.42% |
| H03A | Thyroid preparations | 84.82% | N06A | Antidepressants | 20.88% |
| H05A | Parathyroid hormones and analogues | 66.33% | C10A | Lipid modifying agents, plain | 17.05% |
| A11C* | Vitamin a and d, incl. combinations of the two | 39.42% | N03A | Antiepileptics | 15.61% |
| N02B | Other analgesics and antipyretics | 37.58% | C09C | Angiotensin ii antagonists, plain | 10.38% |
| A01A | Stomatological preparations | 23.05% | L02B | Hormone antagonists and related agents | 4.22% |
| A12A | Calcium | 21.59% | | | |
| N06A* | Antidepressants | 20.88% | | | |
| C07A | Beta blocking agents | 20.81% | | | |

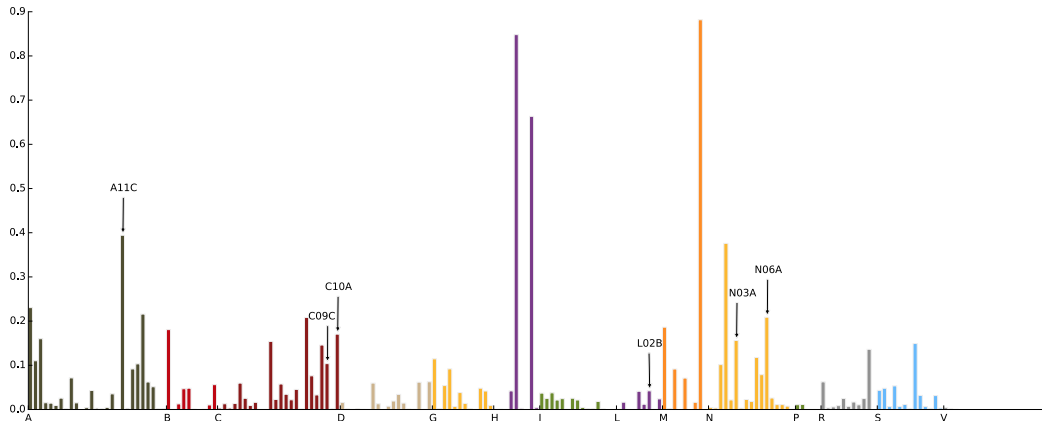

Medication predictions for a simpler patient. Note that the high-prediction medications are clinically reasonable given the billing codes in the sequence. Figure representation as in case 1.

CASE 3.

| ICD-9 code | Code description | Time estimate (ago) |
|---|---|---|
| 332.0 | Paralysis agitans | 5.0 years ago |
| 332.0 | Paralysis agitans | 4.7 years ago |
| 332.0 | Paralysis agitans | 4.5 years ago |
| 332.0 | Paralysis agitans | 4.0 years ago |
| 332.0 | Paralysis agitans | 3.5 years ago |
| 332.0 | Paralysis agitans | 3.0 years ago |
| 332.0 | Paralysis agitans | 2.7 years ago |
| 332.0 | Paralysis agitans | 2.4 years ago |
| 332.0 | Paralysis agitans | 2.0 years ago |
| 332.0 | Paralysis agitans | 1.7 years ago |
| 332.0 | Paralysis agitans | 1.0 years ago |
| 332.0 | Paralysis agitans | 9.9 months ago |
| 332.0 | Paralysis agitans | 4.1 months ago |
| 332.0 | Paralysis agitans | now |

Predicted vs. actual medication classes for Case 3. Table structure as in case 1.

| | Top predictions | Prob. | | True labels | Prob. |
|---|---|---|---|---|---|
| **N04B** | Dopaminergic agents | 97.66% | **C10A** | Lipid modifying agents, plain | 13.90% |
| **N03A** | Antiepileptics | 34.01% | **C09A** | Ace inhibitors, plain | 9.21% |
| **N02B** | Other analgesics and antipyretics | 32.81% | **C01E** | Other cardiac preparations | 5.56% |
| **N06A** | Antidepressants | 26.10% | **C02C** | Antiadrenergic agents, peripherally acting | 0.72% |
| **N02A** | Opioids | 20.33% | **G03B** | Androgens | 0.32% |
| | | | **A14A** | Anabolic steroids | 0.08% |

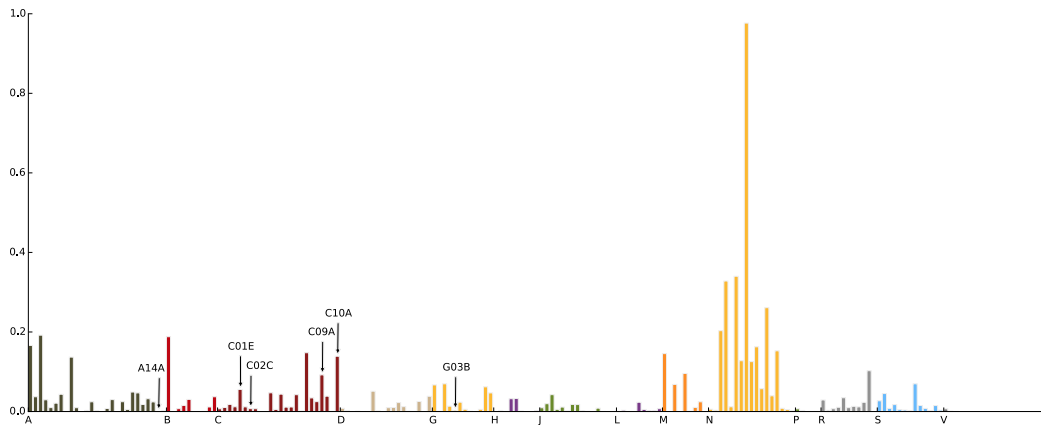

Medication predictions for a patient with only one ICD-9 code, repeated many times over five years. The medications listed under true labels are not indicated for paralysis agitans (Parkinson's disease), but the patient was surely taking them for reasons not documented in the ICD-9 sequence. The model predicted mostly reasonable medications for a patient with Parkinson's disease, especially Dopaminergic agents, which is the primary treatment for the disease. Figure representation as in case 1, above.

