# Peer review of "Predicting Medications from Diagnostic Codes with Recurrent Neural Networks"

_ICLR 2017 — accepted_

[Official Review · AnonReviewer3 · rating 6 · confidence 3 · 16 Dec 2016 (modified: 28 Jan 2017)]
**Good medical application paper for a medical or data science venue**

This is a well-conducted and well-written study on the prediction of medication from diagnostic codes. The authors compared GRUs, LSTMs, feed-forward networks and random forests (making a case for why random forests should be used, instead of SVMs) and analysed the predictions and embeddings.

The authors also did address the questions of the reviewers.

My only negative point is that this work might be more relevant for a data science or medical venue rather than at ICLR.

[Official Review · AnonReviewer2 · rating 7 · confidence 5 · 17 Dec 2016]
**Thorough empirical investigation of an interesting and (to my knowledge) novel application area**

This is a well written, organized, and presented paper that I enjoyed reading.  I commend the authors on their attention to the narrative and the explanations.  While it did not present any new methodology or architecture, it instead addressed an important application of predicting the medications a patient is using, given the record of billing codes.  The dataset they use is impressive and useful and, frankly, more interesting than the typical toy datasets in machine learning.  That said, the investigation of those results was not as deep as I thought it should have been in an empirical/applications paper.  Despite their focus on the application, I was encouraged to see the authors use cutting edge choices (eg Keras, adadelta, etc) in their architecture.  A few points of criticism:

-The numerical results are in my view too brief.  Fig 4 is anecdotal, Fig 5 is essentially a negative result (tSNE is only in some places interpretable), so that leaves Table 1.  I recognize there is only one dataset, but this does not offer a vast amount of empirical evidence and analysis that one might expect out of a paper with no major algorithmic/theoretical advances.  To be clear I don't think this is disqualifying or deeply concerning; I simply found it a bit underwhelming.

- To be constructive, re the results I would recommend removing Fig 5 and replacing that with some more meaningful analysis of performance.  I found Fig 5 to be mostly uninformative, other than as a negative result, which I think can be stated in a sentence rather than in a large figure.

- There is a bit of jargon used and expertise required that may not be familiar to the typical ICLR reader.  I saw that another reviewer suggested perhaps ICLR is not the right venue for this work.  While I certainly see the reviewer's point that a medical or healthcare venue may be more suitable, I do want to cast my vote of keeping this paper here... our community benefits from more thoughtful and in depth applications. Instead I think this can be addressed by tightening up those points of jargon and making the results more easy to evaluate by an ICLR reader (that is, as it stands now researchers without medical experience have to take your results after Table 1 on faith, rather than getting to apply their well-trained quantitative eye). 

Overall, a nice paper.

[Official Review · AnonReviewer1 · rating 8 · confidence 4 · 17 Dec 2016 (modified: 21 Jan 2017)]
**Strong application work, very important problem**

In light of the detailed author responses and further updates to the manuscript, I am raising my score to an 8 and reiterating my support for this paper. I think it will be among the strongest non-traditional applied deep learning work at ICLR and will receive a great deal of interest and attention from attendees.

-----

This paper describes modern deep learning approach to the problem of predicting the medications taken by a patient during a period of time based solely upon the sequence of ICD-9 codes assigned to the patient during that same time period. This problem is formulated as a multilabel sequence classification (in contrast to language modeling, which is multiclass classification). They propose to use standard LSTM and GRU architectures with embedding layers to handle the sparse categorical inputs, similar to that described in related work by Choi, et al. In experiments using a cohort of ~610K patient records, they find that RNN models outperform strong baselines including an MLP and a random forest, as well as a common sense baseline. The differences in performance between the recurrent models and the MLP appear to be large enough to be significant, given the size of the test set.

Strengths:
- Very important problem. As the authors point out, two the value propositions of EHRs -- which have been widely adopted throughout the US due to a combination of legislation and billions of dollars in incentives from the federal government -- included more accurate records and fewer medication mistakes. These two benefits have largely failed to materialize. This seems like a major opportunity for data mining and machine learning.
- Paper is well-written with lucid introduction and motivation, thorough discussion of related work, clear description of experiments and metrics, and interesting qualitative analysis of results.
- Empirical results are solid with a strong win for RNNs over convincing baselines. This is in contrast to some recent related papers, including Lipton & Kale et al, ICLR 2016, where the gap between the RNN and MLP was relatively small, and Choi et al, MLHC 2016, which omitted many obvious baselines.
- Discussion is thorough and thoughtful. The authors are right about the kidney code embedding results: this is a very promising result.

Weaknesses:
- The authors make several unintuitive decisions related to data preprocessing and experimental design, foremost among them the choice NOT to use full patient sequences but instead only truncated patient sequences that each ends at randomly chosen time point. This does not necessarily invalidate their results, but it is somewhat unnatural and the explanation is difficult to follow, reducing the paper's potential impact. It is also reduces the RNN's potential advantage.
- The chosen metrics seem appropriate, but non-experts may have trouble interpreting the absolute and relative performances (beyond the superficial, e.g., RNN score 0.01 more than NN!). The authors should invest some space in explaining (1) what level of performance -- for each metric -- would be necessary for the model to be useful in a real clinical setting and (2) whether the gaps between the various models are "significant" (even in an informal sense).
- The paper proposes nothing novel in terms of methods, which is a serious weakness for a methods conference like ICLR. I think it is strong enough empirically (and sufficiently interesting in application) to warrant acceptance regardless, but there may be things the authors can do to make it more competitive. For example, one potential hypothesis is that higher capacity models are more prone to overfitting noisy targets. Is there some way to investigate this, perhaps by looking at the kinds of errors each model makes?

I have a final comment: as a piece of clinical work, the paper has a huge weakness: the lack of ground truth labels for missing medications. Models are both trained and tested on data with noisy labels. For training, the authors are right that this shouldn't be a huge problem, provided the label noise is random (even class conditional isn't too big of a problem). For testing, though, this seems like it could skew metrics. Further, the assumption that the label noise is not systemic seems very unlikely given that these data are recorded by human clinicians. The cases shown in Appendix C lend some credence to this assertion: for Case 1, 7/26 actual medications received probabilities < 0.5. My hunch is that clinical reviewers would view the paper with great skepticism. The authors will need to get creative about evaluation -- or invest a lot of time/money in labeling data -- to really prove that this works.

For what it is worth, I hope that this paper is accepted as I think it will be of great interest to the ICLR community. However, I am borderline about whether I'd be willing to fight for its acceptance. If the authors can address the reviewers' critiques -- and in particular, dive into the question of overfitting the imperfect labels and provide some insights -- I might be willing to raise my score and lobby for acceptance.

[Final Decision · Program Chairs · 06 Feb 2017]
**ICLR committee final decision**

This paper applies RNNs to predict medications from billing costs. While this paper does not have technical novelty, it is well done and well organized. It demonstrates a creative use of recent models in a very important domain, and I think many people in our community are interested and inspired by well-done applications that branch to socially important domains. Moreover, I think an advantage to accepting it at ICLR is that it gives our "expert" stamp of approval -- I see a lot of questionable / badly applied / antiquated machine learning methods in domain conferences, so I think it would be helpful for those domains to have examples of application papers that are considered sound.